# The Effect of Magnesium Supplementation on Endothelial Function: A Randomised Cross-Over Pilot Study

**DOI:** 10.3390/ijerph18158169

**Published:** 2021-08-02

**Authors:** Jennifer Byrne, Caitríona Murphy, Jennifer B. Keogh, Peter M. Clifton

**Affiliations:** 1Clinical and Health Sciences, University of South Australia, Adelaide, SA 5001, Australia; jenniferbyrne12@hotmail.com (J.B.); caitriona49@hotmail.com (C.M.); jennifer.keogh@unisa.edu.au (J.B.K.); 2School of Biological and Health Sciences, Technological University Dublin, D07 EWV4 Dublin, Ireland

**Keywords:** magnesium, flow mediated dilatation, endothelial function

## Abstract

Evidence supports an association between low magnesium (Mg) intake and coronary heart disease and between Mg intake and endothelial function. The aim of this study was to assess the effect of one week of Mg supplementation on endothelial function, assessed by flow mediated dilatation (FMD). Nineteen healthy men and women completed this cross-over pilot study in which participants were randomised to take an over-the-counter magnesium supplement for one week or to follow their usual diet. Weight, FMD and blood pressure (BP) were taken on completion of each intervention and 24 h urine collections and blood samples were taken to assess compliance. Baseline serum Mg was within normal range for all participants. Urinary Mg and urinary magnesium-creatinine ratio (Mg/Cr) significantly increased between interventions, (*p* = 0.03, *p* = 0.005, respectively). No significant differences in FMD or BP were found between the interventions. A significant negative correlation was seen between age and FMD (r = −0.496, *p* = 0.031). When adjusted for age, saturated fat was negatively associated with FMD (*p* = 0.045). One week of Mg supplementation did not improve FMD in a healthy population.

## 1. Introduction

There is evidence from prospective cohort studies to support an association between Mg intake and cardiovascular disease (CVD) incidence and mortality [1,2,3,4,5,6]. In the US studies whole grain was the main contributor to dietary magnesium, followed by green leafy vegetables, nuts, and legumes however other components of these foods such as fibre and polyphenols may provide the health benefit rather than the magnesium. In the Honolulu Heart Study the excess risk attributable to low dietary magnesium ranged from 1.5- to 1.8-fold after adjustment for both dietary factors and cardiovascular risk factors (*p* < 0.05) [5]. Further evidence suggesting that magnesium itself may be protective comes from studies showing an association between serum magnesium and urinary magnesium and CVD events [3,7,8,9,10,11]. Given that magnesium directly opposes the stimulatory effects of calcium in nerves and smooth and cardiac muscles it is reasonable to postulate that magnesium may directly influence smooth muscle function and indirectly via influencing endothelial function [12]. There is also observational evidence for an association between Mg intake and endothelial function as assessed by flow mediated dilatation (FMD) [13,14,15,16]. Endothelial dysfunction is involved in the initiation of atherosclerosis and is a predictor of CHD [17,18]. Intervention trials investigating the effects of Mg supplementation on FMD have mixed results with improvements in FMD [19,20] in studies with people with coronary artery disease or diabetes but not in others with healthy participants or those with overweight and obesity or metabolic syndrome [21,22,23,24]. These conflicting results suggest that it is not yet possible to draw firm conclusions regarding the effects of Mg supplementation on FMD especially in the absence of clinical disease. If increased magnesium intake decreases clinical disease by improving FMD we should best be able to see it in relatively young healthy people on no medication.

The aim of the present pilot study was to determine the effect of 7 days of Mg supplementation on FMD in comparison to FMD on no Mg supplementation to demonstrate a possible mechanism for the benefit of magnesium-rich foods in healthy people.

## 2. Methods 

### 2.1. Subject Recruitment

Twenty-one participants (4 males, 17 females), aged between 19 and 75 years were recruited through advertisement on University notice boards. Inclusion criteria were age 18–75 years, body mass index (BMI) ≥ 18 kg/m^2^ ≤ 35 kg/m^2^, weight stability in the preceding 6 months, systolic blood pressure (SBP) < 140 mmHg, diastolic blood pressure (DBP) < 90 mmHg. Participants were excluded if taking antihypertensive medication, systemic steroids, folate supplementation, non-steroidal anti-inflammatory drugs or drugs acting on the endothelium, had known metabolic disease such as liver or kidney disease or previous clinical cardiovascular disease.

### 2.2. Study Design

In a crossover design participants were randomised to take an over the counter magnesium supplement (magnesium citrate 927.6 mg, 150 mg of magnesium) two for women and 3 for men equivalent to the Australia’s National Health and Medical Research Council Recommended Daily Intake (RDI) of 400–420 mg/day for men and 310–320 mg/day for women [25] in addition to usual diet for 1 week and then follow their usual diet without supplements for another week. Participants were requested to attend the research clinic at the University of South Australia for an initial screening visit at which eligibility criteria was confirmed and for two other visits in randomised order, one week on normal diet and one week on normal diet with supplements. On both occasions weight, blood pressure and FMD were measured and a blood sample taken. Participants were asked to fast from 10 p.m. the evening before each visit and to avoid alcohol intake. Participants were provided with a digital weighing scales and food diaries and were requested to collect a 3-day weighed food record (two weekdays, one weekend day) at baseline. Participants were requested to collect a 24 h sample of urine on 2 occasions; baseline and on their final day of taking supplements and were provided with suitable collection bottles.

A randomised list to determine whether participants followed normal diet or followed normal diet with supplements on week 1 was generated using http://randomization.com/ (accessed on 30 January 2017). The list was stored in a password protected document on office computers and a hardcopy in a locked cabinet. The researcher taking the FMD measurements was blind to randomisation order. 

### 2.3. Weight and Height

At the initial visit, participants had body height measured to the nearest 0.1 cm with a stadiometer (SECA, Hamburg, Germany) while barefoot. Body weight was measured at each visit to the nearest 0.05 kg by using calibrated electronic digital scales (SECA Hamburg, Germany) while participants wore light clothing and no footwear.

### 2.4. Dietary Intake

Food diaries, collected before any intervention, were analysed using Foodworks Professional Edition 2007 (version 5; Foodworks Professional Edition; Xyris Software, Highgate Hill, Australia) to determine habitual dietary intakes of Mg.

### 2.5. Blood Pressure

Measurements were taken using an automated sphygmomanometer (SureSigns V3; Philips, North Ryde, Australia) with the participant in a seated position. A total of four blood pressure measurements were taken consecutively at visit two and three. The initial reading was discarded and 3 consistent measurements averaged. Consistent measurements defined as <10 mmHg differences between SBP.

### 2.6. FMD

Endothelial function was assessed by endothelium-dependent FMD of the right brachial artery. This was measured in the longitudinal plane above the antecubital fossa with an 8.8-MHz linear array transducer (MySono U6; Samsung Medison) according to published guidelines [26,27]. The brachial artery diameter was measured before and after forearm ischemia caused by inflation of a sphygmomanometer cuff applied to the right forearm 2 cm below the olecranon process to 200 mmHg for 5 min. Continuous longitudinal 2-dimensional images of the brachial artery were obtained and digitally recorded during quiet rest (15 s) (baseline file) and from 15 s before cuff release and during reactive hyperemia (2 min) (deflation file). 

### 2.7. FMD Analysis

Ultrasound images were recorded at a rate of 30 frames/s using screen-capture software (DebutVideo Capture Software Professional V1.82; NCH Software) without QRS gating because previous studies show that continuous recording has good agreement with R-wave gated measures when edge-detection software is used [28]. FMD baseline and deflation video files were coded and stored for blinded offline analysis by two trained individuals using edge-detection software (Brachial Analyzer for Research V6.1.3; Medical Imaging Application LLC, Coralville, IA, USA). A region of interest was defined over a clear section of vessel with care to ensure that the region of interest was the same size and position for both baseline and deflation files. The automated edge-detection feature of the software was used to perform a frame-by-frame analysis to generate artery diameter (mm) values for both baseline and deflation files. Baseline was calculated as the average of 60-s pre-inflation diameter measures and defined as an independent baseline. From the deflation file, a second baseline was derived by calculating the average diameter just after cuff release and the peak diameter was determined as the maximum diameter post cuff release. With the use of this baseline and peak diameter from the deflation file, the absolute change and the percent change was calculated. 

### 2.8. Laboratory Analysis

Blood was collected into two tubes, without additive, from a cannula inserted into the brachial vein. One tube was delivered to an accredited laboratory (SA Pathology, Adelaide, Australia) for measurement of serum magnesium concentration. The other sample was allowed to clot for 30 min and centrifuged at 4000 rpm for 10 min to isolate serum. The serum was aliquoted and stored in a −80 °C freezer. Total weight of the 24-h urine sample was recorded. A small sample (20–30 mL) of the urine was sent to an accredited laboratory (SA Pathology, Adelaide, Australia) for analysis of urinary magnesium and creatinine concentration. The results were multiplied by the weight of the original urine sample to calculate 24-h urinary magnesium and 24-h creatinine. Two 10 mL urine samples were also collected and frozen at −20 °C.

### 2.9. Ethics

This study was approved by the University of South Australia’s Human Research Ethics Committee (ethics approval number: 0000035834). All participants gave written informed consent. This trial was registered at http://www.anzctr.org.au/ (accessed on 30 January 2017) as ACTRN12617000160336.

### 2.10. Statistical Analysis

The primary outcome for the statistical analysis was the effect of Mg supplementation for one week compared with one week without supplementation on FMD. On the basis of power calculations from a previous study using potassium [29] 35 subjects were required to detect a mean difference in FMD of 0.04 mm (1% of baseline diameter) (α = 0.05; 80% power). Statistics were performed using IBM SPSS software (version 21; IBM, Chicago, IL, USA). Significance was set at *p* < 0.05. Data was assessed for normality numerically by analysis of differences between the mean and median, skewness and kurtosis, visually by Q-Q plots and statistically by the Shapiro–Wilks test. Results are expressed as means ± standard deviation (SD) or medians with IQR. Comparisons between baseline and post supplementation parameters were performed by using a paired *t* test or Wilcoxon’s sign-rank statistic as appropriate. Repeated-measures analysis of variance (ANOVA) with condition (i.e., baseline or post supplement) as the within-subject factor was used to assess the effect of supplementation on FMD and blood pressure. Sphericity was assumed as the repeated-measures variable has only two levels. Weight, age, baseline SBP and DBP, baseline urinary Mg, average dietary Mg intake and an independent baseline measurement of blood vessel diameter were included as covariates. The coefficient of variation (CV%) was calculated to assess the reproducibility of FMD measurement This was carried out by dividing the SD of the difference between control phase and intervention FMD by the average of the means of the control phase and intervention FMD and multiplying by 100. Pearson’s correlation was used to assess association between FMD, age, weight, BMI and dietary variables. A backwards linear regression was used to explore the association between FMD and these variables with adjustment for specific variables. Carryover was assessed by adding treatment order as a factor.

## 3. Results

Nineteen participants completed the study (Table 1). Dietary analysis is shown in Table 2. 

### 3.1. Effect of Supplementation on Magnesium Status 

The increase in urinary Mg following supplementation approached significance (Table 3). There was a significant rise in the Mg/Cr ratio. Two participants were excluded from the analysis on the basis that creatinine excretion was below the normal range for gender in either the control phase or intervention urine collection, raising a query over compliance with completeness of collection of the sample. A second Wilcoxon-Sign-Rank Test was carried out with these participants excluded and indicated a significant rise in both urinary Mg and Mg/Cr ratio.

### 3.2. Primary and Secondary Endpoints

There was no difference between the magnesium supplementation phase and control phase on the absolute change in FMD and no difference in the mean percentage change in FMD from the control phase to intervention phase (Table 4). The co-variants age, weight, baseline SBP and DBP, baseline serum Mg, baseline urinary Mg, average dietary Mg intake, average dietary calcium intake, dietary Ca/Mg and an independent baseline measurement of blood vessel diameter were included in the analysis, all of which had no effect. There were no significant differences in SBP, DBP and mean arterial pressure (MAP) from the control phase to the intervention phase. For SBP, DBP and MAP, the co-variants weight, baseline serum Mg, baseline urinary Mg, average dietary Mg intake, average dietary calcium intake, dietary Ca/Mg and an independent baseline measurement of blood vessel diameter were included in the analysis. In each case, findings remained non-significant. Treatment order had no effect.

There was no significant difference in weight from the control phase to the intervention phase.

As there were no significant differences between FMD from the control phase and intervention phase an average FMD was computed and used in correlational analysis. The two FMD measures were highly correlated (r = 0.6). Age was significantly negatively correlated with FMD (r = −0.5) (Figure 1), Weight, BMI (r = 0.1), dietary Mg, dietary calcium, dietary Ca/Mg ratio, dietary potassium, dietary sodium, dietary saturated fat (r = 0.1), dietary polyunsaturated fat (PUFA) (fats expressed as % of energy) did not correlate with FMD on univariate analysis.
**Correlation Coefficient**−0.496**Significance***p* = 0.031

Age accounted for 20% of variance in FMD. BMI and saturated fat intake were related to FMD when adjusted for age. BMI showed a positive association (*p* = 0.034), and saturated fat (% energy) a negative association with FMD (*p* = 0.045). These 3 variables together accounted for about 40% of the variance in FMD.

## 4. Discussion

The primary finding of this study is that one week of magnesium supplementation equivalent to the RDI resulted in no difference in FMD or BP in a healthy population. This is similar to several [21,22,23,24] but not all [19,20] previous studies. The positive studies [19,20] had people with frank disease and low baseline FMD whereas the negative studies included people with no disease or just risk factors for disease. Populations with low Mg may respond better to Mg supplementation and hence see greater benefits on FMD [23]. Our population, apart from 2 individuals had a baseline 24 h Mg that was in the replete range (>3.4 mmol/24 h) Although the study was underpowered to see a positive significant effect of magnesium on FMD in fact the opposite was seen suggesting no benefit of magnesium at all and which was not confounded by low numbers.

Systematic reviews and meta-analyses have provided modest support for an association between Mg supplementation and an improvement in BP [31,32,33]. Our findings are consistent with Cosaro [21] who investigated Mg supplementation of a similar amount for 8 weeks in a healthy population with baseline Mg status in the normal range. Similarly Joris et al. [34] observed no improvements in BP in overweight and obese subjects following Mg supplementation. Other studies have observed improvements in blood pressure among participants with higher baseline blood pressure than in the present study [35]. In our analysis, the strongest dietary predictor of FMD was saturated fat, as after adjustment for age and BMI, a negative association with FMD was observed. BMI only became significant after adding saturated fat intake and age suggesting that part of the reason BMI may be positively associated with FMD may be due its correlation with saturated fat and age and adjusting for these reveals the unexpected positive association. Although in a cross-sectional study larger visceral adipocytes were associated with lower FMD there was no association between BMI and visceral adipose size [36]. Thus an increased BMI may not necessarily have an adverse effect. This finding with fat is consistent with other studies in the literature, which have explored the effect of fat on FMD [37,38].

It is a strength of the study that FMD measurements were taken by a single trained operator who was blind to the randomisation order. In addition the FMD recordings were coded to enable blinded analysis of the recordings. Continuous diameter measurement and automated edge-detection software were used in our study and there was a strong correlation between the FMD measures on both occasions This was a healthy population sample which may limit the potential to see improvements in FMD and blood pressure. There was no placebo, so participants knew when they were receiving the intervention. No other markers of endothelial function were taken. The duration of supplementation in the study was relatively short. This duration was chosen because our previous work has shown that FMD improves after 2 days when sodium is reduced or potassium increased [39] and there is evidence from animal studies that plasma and cellular equilibration occurs over 2–3 h [12].

### Limitations

The small sample size meant that if a positive effect of magnesium was present we may not have had enough power to see a statistically significant effect. As it transpired there was no apparent effect of magnesium as FMD was greater after the no supplementation period. For this finding to be significant over 300 volunteers would be required such is the variance in the change in FMD. Another limitation was that the volunteers were young healthy normal weight people so that our conclusions as to the lack of effect of magnesium apply only to this population. A recent metanalysis confirms the lack of effect of magnesium in young healthy people [40]).

## 5. Conclusions

In our study, no significant differences in FMD were seen following one week of Mg supplementation supporting the majority of the interventions examining FMD. Different endpoints other than FMD need to be measured, such as inflammatory markers, to elucidate the potential benefit of magnesium rich foods. There would appear to be little value in performing additional studies of FMD in healthy volunteers.

## Figures and Tables

**Figure 1 ijerph-18-08169-f001:**
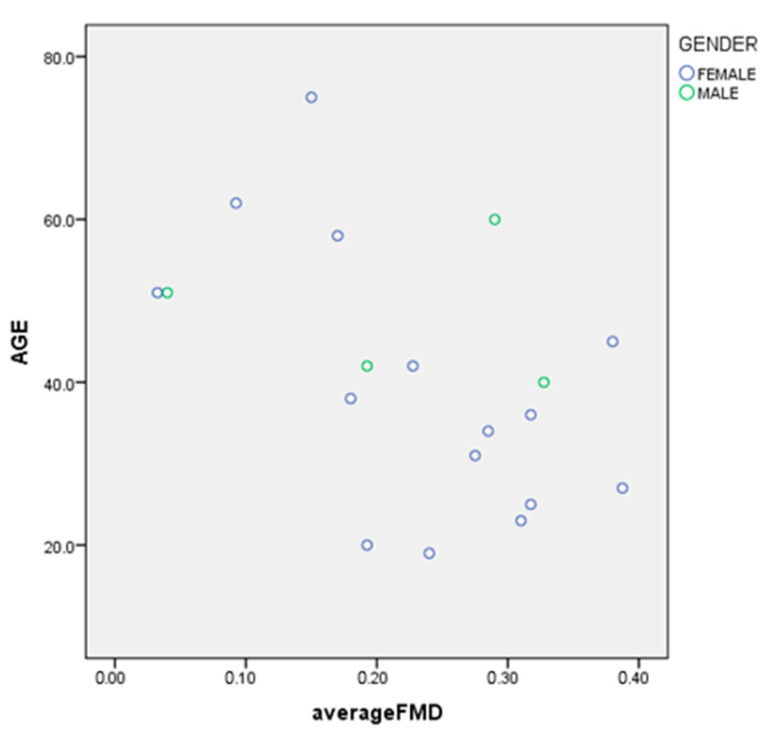
Age and FMD.

**Table 1 ijerph-18-08169-t001:** Baseline characteristics.

	Minimum	Maximum	Mean/Median	SD/IQR
Age (years)	19	75	39	16.04
Weight (kg)	45.04	98.51	59.3 (median)	14.40 (IQR)
BMI (kg/m^2^)	17.9	31.7	21.2 (median)	5.59 (IQR)
SBP(mmHg)	93	120	109	7.29
DBP (mmHg)	55	80	67	7.30
MAP	66	93	77	7.17
Serum Mg (mmol/L)	0.76	0.94	0.85	0.05
Urinary Mg (mmol/24 h)	2.03	5.23	3.23 (median)	1.33 (IQR)
Urinary Cr (mmol/24 h)	3.64	20	8.82 (median)	3.19 (IQR)
Mg/Cr Ratio	0.19	0.70	0.38 (median)	0.17 (IQR)

Key: Body mass index (BMI), systolic blood pressure (SBP), diastolic blood pressure (DBP), mean arterial pressure (MAP), magnesium (Mg), creatinine (Cr), interquartile range (IQR). Normal range serum Mg (0.7–1.0 mmol/L). Normal ranges for creatinine excretion: males: 9–19 mmol/24 h, females: 6–13 mmol/24 h.

**Table 2 ijerph-18-08169-t002:** Dietary Intake.

Dietary Intake mg/day (mmol/day)	Minimum	Maximum	Mean/Median	Std. Deviation/IQR
Mg intake	138 (5.68)	510 (20.99)	339 (13.95)	106.75
Male Mg intake (N = 4)	199 (8.19)	429 (17.65)	328 (13.50)	51.03
Female Mg intake (N = 15)	138 (5.68)	510 (20.99)	343 (14.12)	97.76
Overall Ca intake	328	1808	782 (median)	580 (IQR)
Male Ca intake	416	1242	1170 (median)	649 (IQR)
Female Ca intake	328	1808	726 (median)	447 (IQR)
Ca/Mg Ratio	0.92	5.26	2.21 (median)	1.74 (IQR)
Sodium	769	4409	1672 (median)	730 (IQR)
Potassium	1325	4388	2887	790
Saturated fat (% energy)	5.44	18.17	11.55	3.26
PUFA fat (% energy)	1.94	8.64	5.33	2.04

Key: magnesium (Mg), calcium (Ca), polyunsaturated fat (PUFA), interquartile range (IQR).

**Table 3 ijerph-18-08169-t003:** Markers of magnesium status from baseline to post-supplementation.

	Serum Mg mmol/L (SD)	Urinary Mg mmol/24 (SD)	Urinary Mg/Cr Ratio	Urinary Cr Excretion mmol/24 h(SD)
N = 19				
Control	0.85 (0.05)	3.39 (0.86)	0.3936 (0.13)	9.34 (3.56)
Intervention	0.86 (0.05)	4.26 (1.65)	0.4868 (0.10)	8.84 (3.05)
Significance	*p* = 0.19	*p* = 0.064	*p* = 0.009	*p* = 0.059
N = 17				
Control	0.085 (0.05)	3.51 (0.86)	0.3655 (0.11)	10.13 (3.37)
Intervention	0.087 (0.06)	4.48 (1.61)	0.4729 (0.09)	9.41 (2.68)
Significance	*p* = 0.2	*p* = 0.03	*p* = 0.005	*p* = 0.102

Abbreviations: magnesium (Mg), creatinine (Cr), magnesium-creatinine ratio (Mg/Cr). Normal ranges for creatinine excretion: males: 9–19 mmol/24 h, females: 6–13 mmol/24 h [30].

**Table 4 ijerph-18-08169-t004:** Primary and Secondary Endpoints.

	Change in FMD (mm) (SD)	Percent Change in FMD (%) (SD)	SBP (mmHg) (SD)	DBP (mmHg) (SD)	MAP (mmHg) (SD)
Control	0.24 (0.11)	6.77 (3.38)	109 (7.16)	66 (7.00)	76 (6.49)
Intervention	0.22 (0.12)	6.21 (3.83)	110 (8.70)	66 (7.53)	76 (7.40)
Significance	*p* = 0.3	*p* = 0.4	*p* = 0.5	*p* = 0.3	*p* = 0.9

Abbreviations: flow-mediated dilation (FMD), systolic blood pressure (SBP), diastolic blood pressure (DBP), mean arterial pressure (MAP), standard deviation (SD).

## Data Availability

The data presented in this study are available on request from the corresponding author.

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
