# Peer review of "The Effect of Magnesium Supplementation on Endothelial Function: A Randomised Cross-Over Pilot Study"

_ijerph, 2021, doi:10.3390/ijerph18158169_

Round 1

Reviewer 1 Report

In this resubmission, Byrne and colleagues addressed my concerns only in part. Therefore, their effort is insufficient to allow further consideration of the paper.

The manuscript presents many methodological flaws also in the resubmission. The authors did not provide any comments on the original concerns.

Specifically, the points of the first revision that were not addressed are:

  • Why the authors did not measure baseline FMD at the screening visit before the Mg supplementation? Effects of Mg should have been analyzed using a repeated measures design by comparing post-treatment vs baseline FMD values.
  • There is no evidence of a washout period after the Mg supplementation week. Indeed, participants who started on Mg supplementation could benefit from Mg effects also during the following week
  • The choice of participants was not appropriate. It is well known that older people have reduced FMD due to increasing endothelial dysfunction. Perhaps a narrower age interval would have been more appropriate.
  • How the authors explain the gender imbalance in their cohort?
  • It is well known that FMD is widely influenced by the menstrual cycle phase. This could be addressed by performing measurements on the same subject at the same phase (e.g. every 4 weeks), which is not the case of this paper.
  • Why the authors reported a sample size estimation and after that chose sample size arbitrarily? Moreover, it is quite curios that the cohort of the mentioned pilot study is the same as the present study.

Author Response

We have added the following to limitations to deal with the study population limitations

Another limitation was that the volunteers were young healthy normal weight people so that our conclusions as to the lack of effect of magnesium apply only to this population. A recent meta-analysis confirms the lack of effect of magnesium  in  young healthy people  (40)

  • Why the authors did not measure baseline FMD at the screening visit before the Mg supplementation? Effects of Mg should have been analyzed using a repeated measures design by comparing post-treatment vs baseline FMD values.

This is a standard repeated measures analysis  which compared FMD on Magnesium and FMD not on magnesium. Baseline measures introduces more noise. Assessment for carryover was  made.

  • There is no evidence of a washout period after the Mg supplementation week. Indeed, participants who started on Mg supplementation could benefit from Mg effects also during the following week.
  • No carryover was seen as treatment order had no effect
  • The choice of participants was not appropriate. It is well known that older people have reduced FMD due to increasing endothelial dysfunction. Perhaps a narrower age interval would have been more appropriate.
  • see above
  • How the authors explain the gender imbalance in their cohort?
  • this was a random result of recruitment
  • It is well known that FMD is widely influenced by the menstrual cycle phase. This could be addressed by performing measurements on the same subject at the same phase (e.g. every 4 weeks), which is not the case of this paper.
  • There are papers describing no effect of menstrual cycle on FMD.  Williams et al 2020 in a meta-analysis and systematic review  found a very low certainty that FMD improved in the luteal phase as there was very high variability  in the studies (I2 was 84%)
  • Why the authors reported a sample size estimation and after that chose sample size arbitrarily? Moreover, it is quite curios that the cohort of the mentioned pilot study is the same as the present study.
  • we omitted any mention of pilot study in the new study. We endeavoured to recruit the desired sample size but failed but in the end this had no effect on the result

Reviewer 2 Report

Authors have addressed all the main concerns of reviewers and improved this pilot study further. 

Author Response

no response required

This manuscript is a resubmission of an earlier submission. The following is a list of the peer review reports and author responses from that submission.

Round 1

Reviewer 1 Report

Byrne and colleagues presented the results of a pilot study evaluating the effect of Magnesium (Mg) supplementation on endothelial function, mainly analyzed through flow-mediated dilation (FMD), in healthy subjects. 

While the topic is certainly of interest, the paper presents several unaddressable methodological issues that, in my opinion, prevent further consideration of this work. Specifically:

  • The Introduction focuses on FMD in CHD and CKD, while the study was performed on healthy subjects.
  • Why the authors did not measure baseline FMD at the screening visit before the Mg supplementation? Effects of Mg should have been analyzed using a repeated measures design by comparing post-treatment vs baseline FMD values.
  • There is no evidence of a washout period after the Mg supplementation week. Indeed, participants who started on Mg supplementation could benefit from Mg effects also during the following week
  • What is the amount of Mg++ contained in 927.64 mg magnesium citrate?
  • The choice of participants was not appropriate. It is well known that older people have reduced FMD due to increasing endothelial dysfunction. Perhaps a narrower age interval would have been more appropriate.
  • How do the authors explain the gender imbalance in their cohort?
  • It is well known that FMD is widely influenced by the menstrual cycle phase. This could be addressed by performing measurements on the same subject at the same phase (e.g. every 4 weeks), which is not the case of this paper.
  • Why the authors reported a sample size estimation and after that chose sample size arbitrarily? Moreover, it is quite curious that the cohort of the mentioned pilot study is the same as the present study.
  • Tables 3 and 4 are missing.
  • In table 5, why data on serum Mg of the 17 analyzed subjects were not reported?
  • No correlation coefficients were reported for correlations between FMD, BMI, and saturated fat. Moreover, how the authors explain that FMD increases with increasing BMI?

Reviewer 2 Report

Overall a good pilot study but the purpose, aims and hypotheses tested are not clearly developed. The manuscript could be further improved by 

  1. Is one week supplementation of Mg sufficient to see physiological changes which could be detected via FMD? What was the scientific rationale for one week. Please include in methods.
  2. Table 5 is very confusing. I suggest focus on the 17 participants complete results to avoid confusion 
  3. It would be better to focus results of males and females separately. FMD does show some differences in gender and Mg intakes vary.  

Reviewer 3 Report

This manuscript is well written and organized. However, there are some areas that need attention with reference to establishing the study’s significance, theoretical base, and sample justification

Content

The literature review is inadequate. What is known and not known with synthesis of the literature is needed to substantiate the significance of conducting this study. Here the author merely discuss the significance of the problem of Mg supplementation in reference to its effects on endothelial function but overall how will this help people in general? What will be prevented? Etc… how bad is the problem?

Can possibly add some statistics to substantiate the need for this study. Where is discussion of morbidity and mortality r/t heart disease, how is lack of magnesium related physiologically?

There is also no theory to support this analysis. You can perhaps add biologically based theories to outline the connections between magnesium and endothelial functioning and heart disease and functioning with definitions and descriptions of the major variables/concepts being tested. How does magnesium improve heart function, how does it act on the endothelium? What is the impact on cardiac function?

There needs to be justification or rationales for the inclusion/exclusion criteria. Why this age group, gender? Perhaps more explanation in the introduction of the significance of studying this group of people can be added.

There also needs to be a justification for the sample size. Why was this sample size chosen? Was there a power analysis? Could the findings be due to lack of an adequate sample. This is not discussed.

The discussion merely discuss what other research has done. I would also put how this research adds or does not add to the body of knowledge about this subject.

What are the future research implications? What more needs to be learned? What can other studies do to take your research a step further?

This study also needs a limitations section. If the author can pinpoint the limitations, some of which are outlined in this review, this may also aid with adding in what is missing.

I would publish if the author can tighten this up with revisions to stengthen this manuscript.